# Neural-Progressive Hedging: Enforcing Constraints in Reinforcement Learning with Stochastic Programming

**Supriyo Ghosh** [*1]          **Laura Wynter**[2]          **Shiau Hong Lim**[2]          **Duc Thien Nguyen**[3]

[1]Microsoft Research, Bangalore, India
[2]IBM Research AI, Singapore
[3]Singapore Management University, Singapore

## Abstract

We propose a framework, called neural-progressive hedging (NP), that leverages stochastic programming during the online phase of executing a reinforcement learning (RL) policy. The goal is to ensure feasibility with respect to constraints and risk-based objectives such as conditional value-at-risk (CVaR) during the execution of the policy, using probabilistic models of the state transitions to guide policy adjustments. The framework is particularly amenable to the class of sequential resource allocation problems since feasibility with respect to typical resource constraints cannot be enforced in a scalable manner. The NP framework provides an alternative that adds modest overhead during the online phase. Experimental results demonstrate the efficacy of the NP framework on two continuous real-world tasks: (i) the portfolio optimization problem with liquidity constraints for financial planning, characterized by non-stationary state distributions; and (ii) the dynamic repositioning problem in bike sharing systems, that embodies the class of supply-demand matching problems. We show that the NP framework produces policies that are better than deep RL and other baseline approaches, adapting to non-stationarity, whilst satisfying structural constraints and accommodating risk measures in the resulting policies. Additional benefits of the NP framework are ease of implementation and better explainability of the policies.

## INTRODUCTION

Reinforcement learning (RL) experienced a surge in popularity when deep models demonstrated superior performance in game playing with Deep Q-learning Networks (DQN)

---

*Corresponding author: supriyoghosh@microsoft.com

[Mnih et al., 2013]. The role of RL was cemented when it was used to beat the reigning Go world champion [Silver et al., 2017]. Improvements to deep RL algorithms have abounded, including RL for continuous state and action spaces, with DDPG [Lillicrap et al., 2015], TRPO [Schulman et al., 2015], and PPO [Schulman et al., 2017]. In spite of these advances, the dominance of RL for real-world problems has lagged. We believe that this is due to three shortcomings.

First, RL policies cannot enforce business rules, or constraints during policy execution. Yet, often structural constraints must be respected for a policy to be implementable. Existing methods, such as constrained policy gradient [Achiam et al., 2017] or "safe" RL methods [García and Fernández, 2015] do not prevent constraint violations during policy execution. Moreover, these methods can often be difficult to train and not scalable to large problems. Second, there is a natural trade-off between expected reward and risk. The majority of RL algorithms seek to maximize the expected return. While there have been RL algorithms that optimize for various risk measures, doing so in a scalable manner and under constraints is still challenging. Third, the sample inefficiency of RL has posed an impediment to solving problems where high-fidelity simulators are not available to generate sufficiently large number of sample trajectories. A hope to overcoming this is through the judicious use of models to explore more sparingly the state and action spaces.

We introduce a framework to address these issues for problems with continuous state and action spaces. An unconstrained RL policy is first trained offline. During the online execution phase, a stochastic program (SP) is used to re-optimize the given RL policy under constraints and risk measures over a short-term future trajectory. Once the next action is chosen, the process repeats in a rolling-horizon fashion using updated state information. We call this neural-progressive hedging (NP). During execution time, the NP method aims to exploit the generalization ability of RL to unseen scenarios that are not experienced during training time,

*Accepted for the 38th Conference on Uncertainty in Artificial Intelligence (UAI 2022).*

jointly with the ability of SP to exploit models and enforce scenario-dependent constraints as well as incorporating risk measures. Since the NP framework relies on a model-based online phase, it is most useful in problem settings where closed-form models of state transitions are relatively good approximations to the true state transitions. Sequential and dynamic resource allocation problems are a key example in which the NP framework excels.

Empirically, we show that the NP method results in policies that offer substantial improvements in reward under various risk measures whilst satisfying hard constraints[1]. Moreover, we observe that the NP policy, with its more sample-efficient initial RL policy followed by the online fine-tuning phase, is able to outperform the fully-trained (and data-hungry) RL policy. An additional benefit of the framework is ease of implementation: it can be implemented using existing deep RL algorithms and off-the-shelf optimization packages. Furthermore, the SP counterpart of the NP method allows easily performing sensitivity analysis on the model parameters and ensures that the business rules or constraints are enforced during online execution of a policy which adds significant transparency and explainability to the policy choices in comparison to deep RL. To that end, the key contributions of the paper are as follows:

1. We define a novel method, neural-progressive hedging, combining stochastic programming model-based online planning with offline, deep RL for a continuous policy that satisfies hard constraints during execution;

2. We incorporate risk-measures such as CVaR without sacrificing model structure or decomposition algorithm; and

3. We demonstrate the efficacy of the NP method on the class of resource allocation models, including two real-world problems: (i) liquidity-constrained portfolio optimization with a CVaR objective; and (ii) dynamic repositioning in a bike sharing system, where the NP method outperforms deep RL, both constrained and unconstrained as well as other baselines.

## PRELIMINARIES

Consider the problem of learning a deterministic policy $\pi : \mathcal{S} \to \mathcal{A}$ in a Markov Decision Process (MDP) given by $(\mathcal{S}, \mathcal{A}, p, f, \gamma, T, G)$, with continuous states $s \in \mathcal{S}$, continuous actions $x \in \mathcal{A}$, transition probability distribution $p(s_{t+1}|s_t, x_t)$, cost function $f(s_t, x_t, s_{t+1}) \in \mathbb{R}$, discount factor $\gamma \in [0, 1]$, decision horizon $T$, and constraint set $G$. We allow $T = \infty$ whenever $\gamma < 1$. The constraint set $G$ contains a set of $K$ additional cost functions $g_1 \ldots g_K$ where $g_k(s_t, x_t, s_{t+1}) \in \mathbb{R}$ and constants $\beta_1 \ldots \beta_K \in \mathbb{R}$.

Our constrained MDP setting follows that of Altman [1999], where we aim to solve the following problem:

$$\underset{\pi}{\text{minimize }} \mathbb{E}_\pi \left[ \sum_{t=1}^{T} \gamma^{t-1} f(s_t, x_t, s_{t+1}) \right] \quad (1)$$

$$\text{s.t. } \mathbb{E}_\pi \left[ \sum_{t=1}^{T} \gamma^{t-1} g_k(s_t, x_t, s_{t+1}) \right] \le \beta_k, \quad k = 1 \ldots K.$$

Without loss of generality we assume a fixed initial state $s_1$. The expectation $\mathbb{E}_\pi$ is taken with respect to randomness induced by the transitions $s_{t+1} \sim p(\cdot|s_t, x_t)$ by taking $x_t = \pi(s_t)$, for all $t$. Problem (1) is very challenging to solve for general MDPs with continuous states and actions. We shall now put this constrained MDP in the context of stochastic programming (SP) from which we borrow many of the algorithmic tools in this work.

The key assumption from SP is that all the randomness or uncertainty in the system comes from external sources. This decoupling of randomness allows us to employ powerful optimization tools in solving the main problem. Assume $T$ is finite and let $\xi_1 \ldots \xi_T$ be random variables such that the next state $s_{t+1}$ is given by $s_{t+1} = \tilde{p}(s_t, x_t, \xi_t)$ where $\tilde{p}$ is a deterministic function once $\xi_t$ is fixed. We call each realization of $\xi = (\xi_1 \ldots \xi_T)$ a *scenario*. Given a particular scenario $\xi$ [2], one can find the best action sequence in "hindsight" by solving the following problem:

$$\underset{x=(x_1 \ldots x_T)}{\text{minimize}} \quad \tilde{f}(x, \xi) \quad (2)$$

$$\text{s.t.} \quad \tilde{g}_k(x, \xi) \le \beta_k, \quad k = 1 \ldots K$$

where we define

$$\tilde{f}(x, \xi) := \sum_{t=1}^{T} \gamma^{t-1} f[s_t, x_t, \tilde{p}(s_t, x_t, \xi_t)]$$

$$\tilde{g}_k(x, \xi) := \sum_{t=1}^{T} \gamma^{t-1} g_k[s_t, x_t, \tilde{p}(s_t, x_t, \xi_t)].$$

If, for each $\xi$, the functions $\tilde{f}$ and $\tilde{g}_k$ for all $k$ are all convex in $x$, then each scenario sub-problem can be readily solved using existing convex optimization tools. To simplify notation, we define the constraint set $\mathcal{G}(\xi) := \{x|\tilde{g}_k(x, \xi) \le \beta_k, k = 1 \ldots K\}$, so problem (2) can be stated simply as $\text{minimize}_{x \in \mathcal{G}(\xi)} \tilde{f}(x, \xi)$.

Suppose that one starts with a finite set $\Xi$ of scenarios, with known probability distribution $q(\xi)$ where $\sum_{\xi \in \Xi} q(\xi) = 1$. One can solve problem (2) for each individual $\xi \in \Xi$ to obtain a mapping $x(\cdot)$ that provides a solution $x(\xi) = (x_1(\xi) \ldots x_T(\xi))$ for each $\xi \in \Xi$. Suppose that the action space $\mathcal{A} \subseteq \mathbb{R}^n$ and $|\Xi| = N$, then $x(\cdot) \in \mathcal{A}^{N \times T} \subseteq \mathbb{R}^{N \times T \times n}$. How could we then reconcile the various $x_t(\xi)$

---

[1]Source codes of our NP method are available here: https://github.com/supriyog/neural-progressive-hedging

[2]We abuse notation slightly by using $\xi$ to refer to both the random variable and its particular realizations.

across all $\xi \in \Xi$, at time $t$? For the resulting solutions to be implementable, one needs to enforce a *nonanticipative* property which states that $x_t$ must only depend on information available at time $t$. From an MDP point of view, the state $s_t$ captures all observations available up to time $t$, represented by $\xi_1 \ldots \xi_{t-1}$, and therefore $x_t$ must only depend on these if it is to be implementable, i.e., $x_t(\xi) = x_t(\xi_1, \ldots, \xi_{t-1})$ and $x_1(\xi)$ must be the same for all $\xi$. All solutions $x(\cdot)$ that satisfy this nonanticipative property can be expressed as:

$$x(\xi) = (x_1, x_2(\xi_1), \ldots, x_T(\xi_1, \ldots, \xi_{T-1})), \quad \forall \xi \in \Xi.$$

We use $\mathcal{M}$ to denote the space of all nonanticipative mappings. Define an inner product on $\mathcal{A}^{N \times T}$ by $\langle x(\cdot), w(\cdot) \rangle := \sum_\xi q(\xi) \sum_{t=1}^T \langle x_t(\xi), w_t(\xi) \rangle$ where $\langle x_t(\xi), w_t(\xi) \rangle$ is the standard inner product in $\mathbb{R}^n$. Given any $\hat{x}(\cdot) \in \mathcal{A}^{N \times T}$, one can find a nonanticipative version $x(\cdot) = P_\mathcal{M}[\hat{x}(\cdot)]$ where $P_\mathcal{M}$ is the orthogonal projection onto $\mathcal{M}$ given by the conditional expectation $x_t(\xi) = \mathbb{E}_{\xi | \xi_1 \ldots \xi_{t-1}} \hat{x}_t(\xi)$ for all $t$ and $\xi$. Note that $P_\mathcal{M}$ can be computed via simple averaging over the appropriate subsets of scenarios.

Define $\mathcal{G} \subseteq \mathcal{A}^{N \times T}$ such that $x(\cdot) \in \mathcal{G}$ iff $x(\xi) \in \mathcal{G}(\xi)$ for all $\xi$. We then aim to solve the following global problem:

$$\underset{x(\cdot) \in \mathcal{G} \cap \mathcal{M}}{\text{minimize}} \quad \mathbb{E}_\xi \tilde{f}(x(\xi), \xi) \tag{3}$$

where $\mathbb{E}_\xi \tilde{f}(x(\xi), \xi) = \sum_{\xi \in \Xi} q(\xi) \tilde{f}(x(\xi), \xi)$. Without the constraint $x(\cdot) \in \mathcal{M}$, problem (3) would in fact be separable and could be decomposed into solving individual scenarios as in problem (2). This problem, however, can still be solved in an iterative manner where each iteration involves solving a slightly modified version of problem (2) for each scenario. This "progressive hedging" algorithm by Rockafellar and Wets [1991], which is an application of the proximal point algorithm, involves keeping track of the solution $x^i(\cdot)$ as well as a Lagrange multiplier $\lambda^i(\cdot)$ in each iteration $i$, until convergence. It also involves a parameter $\nu^i > 0$, which may be constant for all $i$. Each iteration involves solving the following steps:

1. At iteration $i$, solve the following for each scenario $\xi$:

$$\hat{x}^i(\xi) \in \arg \min_{x(\xi) \in \mathcal{G}(\xi)} \quad \tilde{f}(x(\xi), \xi) + \langle \lambda^i(\xi), x(\xi) \rangle$$
$$+ \frac{\nu^i}{2} \|x(\xi) - x^i(\xi)\|^2 \tag{4}$$

2. Compute $x^{i+1}(\cdot) = P_\mathcal{M}[\hat{x}^i(\cdot)]$.

3. Update the Lagrange multiplier $\lambda^{i+1}(\cdot) = \lambda^i(\cdot) + \nu^i[\hat{x}^i(\cdot) - x^{i+1}(\cdot)]$.

In the case where $\tilde{f}$ and $\mathcal{G}$ are both convex, the algorithm is guaranteed to converge to an optimal solution $x^*(\cdot)$ of problem (3) starting from arbitrary $x^1(\cdot)$ and $\lambda^1(\cdot)$. Local convergence to a stationary point for nonconvex $\tilde{f}$ was shown by Rockafellar [2019].

---

**Algorithm 1** Neural-Progressive Hedging Algorithm

**Initialization:** Obtain RL policy $\pi_\theta$. Define inner convergence criterion $\epsilon$, convex combination parameters $\kappa^i$ and penalty parameters $\nu^i > 0$ for $i > 0$.
**for** $\tau = 1, 2, \ldots,$ **do**
  Observe state $s_{(\tau)}$. Sample scenario set $\Xi$, and query $\pi_\theta$ to obtain $x^\pi(\cdot)$. Set $x^1(\cdot) = x^\pi(\cdot)$. Set $\lambda^1(\cdot) = 0$, $u^1(\cdot) = 0$ and $i = 1$.
  **while** convergence criterion $\delta^i > \epsilon$ **do**
    **1.** Solve, for each $\xi \in \Xi$, (5) (or (4) for the risk-neutral case) to obtain $\hat{x}^i(\xi)$ and $\hat{y}^i(\xi)$.
    **2.** Set $x^{i+1}(\cdot) = \kappa^i x^\pi(\cdot) + (1 - \kappa^i) P_\mathcal{M}[\hat{x}^i(\cdot)]$. Set $y^{i+1}(\cdot) = \mathbb{E}_\xi[\hat{y}^i(\cdot)]$.
    **3.** Update multipliers: $\lambda^{i+1}(\cdot) = \lambda^i(\cdot) + \nu^i(\hat{x}^i(\cdot) - x^{i+1}(\cdot))$ and $u^{i+1}(\cdot) = u^i(\cdot) + \nu^i(\hat{y}^i(\cdot) - y^{i+1}(\cdot))$.
    **4.** Update $\kappa^i, \nu^i$.
    **5.** Convergence test: $\delta^{i+1} := \|\hat{x}^i(\cdot) - x^i(\cdot)\| + \|\hat{y}^i(\cdot) - y^i(\cdot)\|$
    **6.** Set $i \leftarrow i + 1$, continue.
  **end while**
  From converged solution $x^*(\cdot)$, obtain and execute $x_1^*$.
**end for**

---

The SP framework can be adapted to measures of risk. Consider CVaR, the conditional value-at-risk, a popular measure for finding risk-averse solutions. The CVaR of a random variable $Z$ at level $\alpha \in [0, 1)$ can be written as:

$$\text{CVaR}_\alpha(Z) := \min_{y \in \mathbb{R}} \left\{ y + \frac{1}{1 - \alpha} \mathbb{E}_Z [\max\{0, Z - y\}] \right\}.$$

CVaR at $\alpha = 0$ gives the expectation. We solve the CVaR version of problem (3), replacing the expectation $\mathbb{E}_\xi$ with $\text{CVaR}_\alpha$, by following a modified progressive hedging algorithm [Rockafellar, 2018] with an introduction of an additional variable $y^i(\xi) \in \mathbb{R}$ and the corresponding dual $u^i(\xi) \in \mathbb{R}$ for each $\xi$. Instead of equation (4), we solve equation (5) in step 1 with corresponding changes in steps 2 and 3.

$$(\hat{y}^i(\xi), \hat{x}^i(\xi)) \in \arg \min_{y(\xi) \in \mathbb{R}, x(\xi) \in \mathcal{G}(\xi)} \left\{ y(\xi) + \frac{1}{1 - \alpha} \cdot \right.$$
$$\max\{0, \tilde{f}(x(\xi), \xi) - y(\xi)\} + \frac{\nu^i}{2} |y(\xi) - y^i(\xi)|^2 +$$
$$\left. u^i(\xi) y(\xi) + \langle \lambda^i(\xi), x(\xi) \rangle + \frac{\nu^i}{2} \|x(\xi) - x^i(\xi)\|^2 \right\} \tag{5}$$

## NEURAL-PROGRESSIVE HEDGING

We introduce Neural-Progressive Hedging (NP) method combining the generalization capability of offline RL with the ability of SP through an online phase to exploit models while enforcing scenario-dependent constraints and risk

measures. The key steps of the NP method are shown compactly in Algorithm 1.

The NP method works as follows: an unconstrained RL policy $\pi_\theta$, parameterized by $\theta$, is obtained by solving (1), or its risk-aware counterpart, without constraints. In each time-step $\tau$, the NP method observes current state $s_{(\tau)}$ and queries RL policy $\pi_\theta$ to get initial action $x^\pi(\cdot)$. The new NP policy is guided by the initial RL policy via a convex combination parameter $\kappa$ so that, at convergence, the executed actions satisfy constraints of $\mathcal{G}$ and the risk measures. Inner iterations are denoted by $i = 1, \ldots$; at each iteration $i$, the SP sub-problems are solved for each scenario $\xi \in \Xi$ with updated Lagrangian multipliers $\lambda^i, u^i$ to obtain the dual solution $\hat{x}^i(\xi)$ and $\hat{y}^i(\xi)$. Then, we project $\hat{x}^i(\xi)$ onto a feasible space $P_\mathcal{M}[\hat{x}^i(\cdot)]$, that satisfies the *nonanticipative* property, by averaging over all the scenarios. The primal solution $x^{i+1}(\cdot)$ is obtained as a convex combination with the initial RL policy $x^\pi(\cdot)$ then projected with $P_\mathcal{M}[\hat{x}^i(\cdot)]$. We then update multipliers $\lambda^i, u^i$, and parameters $\kappa^i$, and $\nu^i$. This iterative process continues until the difference between primal and dual solutions, $\delta^i$, is below a pre-defined threshold $\epsilon$.

**Resource Allocation Problems:** The NP approach is particularly effective for the class of resource allocation problems. In such applications, the main source of uncertainty is external – consider stock price changes or customer demands – and to a large extent such random variables are unaffected by the actions of the policy. A scenario generator can hence be readily trained using historical data. The set of scenarios, $\Xi$, is obtained by sampling from such a scenario generator. Given a scenario, $\xi$, this policy can then be queried at any state $s_t$ to obtain the corresponding action $x_t$. Given a finite scenario set $\Xi$, we can obtain from $\pi_\theta$ its solution $x^\pi(\cdot) \in \mathcal{M}$.

## THEORETICAL ANALYSIS

The parameter $\kappa^i$ blends the offline RL policy with the solution from SP (Step 2 in Algorithm 1). The assumption below covers the settings of warm start, where $\kappa^1 = 1$ and $\hat{\imath} = 2$, and imitation learning, where $\kappa^i$ is a decreasing sequence such as $(1 + i)^{-2}$, where $1 \leq \hat{\imath} < \infty$.

**Assumption 1 (Imitation learning and warm start)** *Let $\kappa^i \to 0$ as $i \to \infty$. Furthermore, there exists an $\hat{\imath}$ such that for all $i \geq \hat{\imath}$, $\kappa^i = 0$.*

**Assumption 2 (Existence and local convexity)** *Assume that the solution set of equation (5) for a CVaR objective, or equation (4) otherwise, is nonempty and finite, $\mathcal{G}(\xi)$ is convex and compact, the gradients of $\tilde{f}$ are locally Lipschitz for each $\xi$ and that the dual penalty parameters $\nu^i$ are sufficiently large for all $i$.*

**Lemma 1** *Under Assumption 1, the NP algorithm is equivalent to the progressive hedging algorithm over an infinite number of iterations.*

**Proof:** Assumption 1 states that there exists a finite iterate $\hat{\imath}$ such that for all $i \geq \hat{\imath}$, $\kappa^i = 0$. Since $x^{i+1}(\cdot) = \kappa^i x^\pi(\cdot) + (1 - \kappa^i) P_\mathcal{M}[\hat{x}^i(\cdot)]$, for all $i' \geq \hat{\imath}$, $x^{i'}(\cdot) = P_\mathcal{M}[\hat{x}^{i'}(\cdot)]$, and hence the update of the primal variable of the algorithm reduces to the progressive hedging update. ∎

Instances of stochastic programming typically make use of discretized support $\Xi$. We thus define the problem (3) in terms of a discrete $\Xi$ and refer to this problem for the remainder of this section.

**Assumption 3 (Discrete support)** *Let $\Xi$ be a discrete support and let $1 \ldots K$ index each scenario corresponding to a random variable $\xi \in \Xi$, with probability $p_k = 1/K$. Then, problem (3) can be expressed as:*

$$\min_{x_k \in \mathcal{G}_k; x_k \in \mathcal{M}} \frac{1}{K} \sum_{k=1\ldots K} \tilde{f}_k(x_k). \tag{6}$$

**Theorem 1 (Convergence of Alg. 1 for Convex $\tilde{f}$)**
*Under Assumptions 1, 2 and 3, along with the convexity of $\tilde{f}$, the sequence of iterates $(x^i(\cdot), y^i(\cdot), \lambda^i(\cdot), u^i(\cdot))$ generated by the NP algorithm is such that*

$$\|x^{i+1} - x^i\|^2 + \|y^{i+1} - y^i\|^2 + (1/\nu^2)\|\lambda^{i+1} - \lambda^i\|^2 + (1/\nu^2)\|u^{i+1} - u^i\|^2 < \|x^i - x^{i-1}\|^2 + \|y^i - y^{i-1}\|^2 + (1/\nu^2)\|\lambda^i - \lambda^{i-1}\|^2 + (1/\nu^2)\|u^i - u^{i-1}\|^2, \text{ and}$$

$$|x^{i+1} - x^*|^2 + |y^{i+1} - y^*|^2 + (1/\nu^2)\|\lambda^{i+1} - \lambda^*\|^2 + (1/\nu^2)\|u^{i+1} - u^*\|^2 < |x^i - x^*|^2 + |y^i - y^*|^2 + (1/\nu^2)\|\lambda^i - \lambda^*\|^2 + (1/\nu^2)\|u^i - u^*\|^2$$

*with equality at $(x^*(\cdot), y^*)$ in the case of finite convergence, and thus converges to a local solution $(x^*(\cdot), y^*)$ with $(\lambda^*(\cdot), u^*(\cdot))$ as $i \to \infty$.*

**Proof:** From Lemma 1, Algorithm 1 is equivalent to the Progressive Hedging Algorithm of Rockafellar [2019] when run for an infinite number of iterations. The convergence of the Progressive Hedging Algorithm to a solution $(x^*(\cdot), y^*(\cdot))$ is thus guaranteed under Assumptions 2 and 3 along with the convexity of $\tilde{f}$. ∎

**Theorem 2 (Convergence of Alg. 1 for Nonconvex $\tilde{f}$)**
*Let Assumptions 1, 2 and 3, hold and let $(x^i(\cdot), y^i(\cdot))$ be a locally optimal solution to each subproblem (5). If sequences $\{x^i, y^i, \lambda^i, u^i\}$ converge to point $\{x^*, y^*, \lambda^*, u^*\}$, then $(x^*(\cdot), y^*(\cdot))$ generated by the NP algorithm is a locally optimal solution to problem (3).*

**Proof:** From Lemma 1, Algorithm 1 is equivalent to the Progressive Hedging Algorithm of Rockafellar [2019] when run for an infinite number of iterations. For nonconvex $\tilde{f}$, when the Progressive Hedging Algorithm converges to a point, under Assumptions 2 and 3, it was shown in Rockafellar and Wets [1991] that the point is a stationary point of the problem (3). ∎

The NP algorithm uses a decomposition of the measurability constraints on the scenario tree from the scenario-specific constraints, and then proceeds to solve the SP by standard optimization methods. It should be noted however that the structure and theoretical properties of the NP hold equally with sample average approximation [Bertsimas et al., 2018].

When combining the unconstrained policy $x^\pi(\cdot)$ with the constrained solution $P_{\mathcal{M}}(\hat{x}^i(\cdot))$, we also show how the quality of $x^{i+1}$ evolves as a function of $x^\pi(\cdot)$ and $P_{\mathcal{M}}(\hat{x}^i(\cdot))$.

**Proposition 1** *Let $\tilde{f}$ be Lipschitz $\forall \xi$, i.e., $\|\tilde{f}(x(\xi),\xi) - \tilde{f}(x'(\xi),\xi)\| \leq L\|x(\xi) - x'(\xi)\|$. We have the following bound as a function of $\kappa^i$ and Lipschitz constant $L$:*

$$\mathbb{E}[\tilde{f}(x^{i+1}(\cdot),\cdot)] \leq \mathbb{E}[\tilde{f}(x^\pi(\cdot),\cdot)] + L(1-\kappa^i)\cdot$$
$$\|P_{\mathcal{M}}(\hat{x}^i(\cdot)) - x^\pi(\cdot)\|.$$

**Proof:** For each scenario $\xi$, we have

$$\tilde{f}(x^{i+1}(\xi),\xi) - \tilde{f}(x^\pi(\xi),\xi) \leq L\|x^{i+1}(\xi) - x^\pi(\xi)\|$$
$$\leq L\|\kappa^i x^\pi(\xi) + (1-\kappa^i)\cdot P_{\mathcal{M}}(\hat{x}^i(\xi)) - x^\pi(\xi)\|$$
$$\leq L(1-\kappa^i)\|P_{\mathcal{M}}(\hat{x}^i(\xi)) - x^\pi(\xi)\| \blacksquare$$

Naturally, we expect an unconstrained RL solution to achieve a higher objective value, but the executed solution may include constraint violations and excessive risk. The parameter $\kappa$, thus controls the trade-off between a higher objective value and constraint satisfaction and risk aversion.

## EXPERIMENTAL RESULTS

To evaluate the performance of the proposed neural-progressive hedging (NP) approach, we conduct experiments on two real-world domains where risk measures and constraints are an integral part of implementable policies: (i) *Liquidity management through portfolio optimization* which seeks to optimally reinvest earnings based on the CVaR whilst maintaining sufficient liquidity; and (ii) *Online repositioning* which seeks to dynamically match supply-demand when resources (here, represented by bikes in a bike-sharing system) must be continuously rebalanced to meet changes in demand whilst respecting the station capacity constraints.

We compare performance of NP method with Constrained Policy Optimization (CPO) [Achiam et al., 2017], and Lagrangian-relaxed Proximal Policy Optimization (PPO-L) [Ray et al., 2019]. DDPG [Lillicrap et al., 2015] is used

to solve the unconstrained RL problems. Note that when $\kappa = 1$, the NP approach returns the DDPG solution. Similarly, when $\kappa = 0$, the NP method returns the results of a pure stochastic program (SP), computed using progressive hedging method [Rockafellar, 2019].

**Experiment settings:** We perform all the experiments on Ubuntu 18.04 virtual machines with 32-core CPU, 64 GB of RAM, and a single Nvidia Tesla P100 GPU. The distributed Ray framework and RLlib [Liang et al., 2017] were used for the DDPG method. The pure SP and NP methods with linear and non-linear objective function are solved using IBM ILOG CPLEX 12.9 and IPOPT [Wächter and Biegler, 2006], respectively. The CPO and PPO-L methods are solved using OpenAI safe RL implementation [Ray et al., 2019].

The unconstrained RL policy used as an expert is computed at each time step $t$ using the DDPG algorithm [Lillicrap et al., 2015]. We use a recurrent neural network (RNN) architecture for training the DDPG method with 1 hidden layer consisting of 25 hidden predictor nodes and a tanh nonlinear activation function. In addition, a long short-term memory (LSTM) model is used to represent the RNN architecture with LSTM cell size 256 and maximum sequence length of 20. Parameter values are as follows: the discounting factor $\gamma = 0.99$, minibatch size $b = 50$ and learning rate $lr = 3e^{-5}$. Two state-of-the-art methods are used to compare with the constrained RL policy: (i) Constrained policy optimization [Achiam et al., 2017]; and (b) Proximal policy optimization with a Lagrangian penalty [Ray et al., 2019]. For both we use a neural network with 2 hidden layers, each consisting of 256 hidden nodes with *tanh* nonlinear activation function. The source codes for the constrained benchmark algorithms can be found at https://github.com/openai/safety-starter-agents.

A discretized scenario tree is used in each decision epoch to solve the NP method for the experiments. For the financial planning example, in each decision period $t$, we generate a two layer scenario tree where the first layer consists of a root node and the second layer includes 1000 nodes, giving rise to 1000 scenarios. The interest rates for each of the scenarios are sampled from a multi-dimensional log normal distribution whose mean and covariance matrix are estimated from the training data set of price movements in the S&P500. For the liquidity constraints, we sample 10 liquidity demand processes from a Gaussian distribution with $\mu = 0.025$ and $\sigma = 0.01$, giving rise to 10,000 scenarios in the second layer of the scenario tree. For the bike sharing problem, due to its complex non-linear objective function, we generate a two-layer tree with 200 leaf nodes, giving rise to 200 scenarios. The demand values at stations for each of the scenarios are sampled from a multi-variate normal distribution whose mean and covariance matrix are learnt from 60 days of training demand data [Ghosh et al., 2019].

## LIQUIDITY-CONSTRAINED PORTFOLIO OPTIMIZATION

The liquidity management problem seeks to optimally reinvest earnings in a portfolio based on the CVaR whilst maintaining sufficient liquidity. Too much liquidity means loss of potential returns and too little incurs borrowing costs. Model-based forecasts of the price movements and liquidity process are generally available in practice. The overall problem thus involves computing allocations across a universe of financial instruments, given observed rewards, prices, and model-based forecasts of the price and liquidity processes. We have one risk-free liquid instrument. In each time step a constraint requires that the amount in the liquid account to satisfy forecasted demand. We consider four portfolios, each with nine stocks and one risk-free instrument. The state at time $t$ includes the current allocation, observed price changes and liquidity demands up to time $t$. The action is a vector, $x_t = (x_{t,1}, ..., x_{t,J})$, of allocations across $J$ instruments at time $t$, where $j = 1$ is the liquid asset. Let $\xi_{t,1}$ be the cumulative liquidity requirement and $W_t$ the wealth at the beginning of time $t$. The constraint set is $\mathcal{G}(\xi) := \{x | W_t \cdot x_{t,1} \geq \xi_{t,1}, \sum_j x_{t,j} = 1, t = 1 \ldots T\}$. The liquidity requirement $\ell^t(\xi)$ for time $t$ is sampled from a Gaussian $\ell \sim \mathcal{N}(\mu, \sigma); \mu_\ell = 0.025, \sigma_\ell = 0.01$ and accumulates over time, i.e., $\xi_{t,1} = L^{t-1} + \ell^t$, where $L^{t-1}$ denotes the accumulated realized liquidity requirement.

We use 11 years of S&P500 daily data from 2009–2019. The data from 2009–2016 is used for training the unconstrained RL policy $\pi_\theta$ and price movement model. For the SP and NP, in each time step, we sample 1000 scenarios from a multi-variate log-normal distribution whose parameters are learnt from the training data. Hyperparameter tuning of $\pi_\theta$ is done using data of 2017–2018. Tests are done on two consecutive 30 working day periods in 2019 (Jan 1-Feb 11, and Feb 12-Mar 25). It should be noted that the experiments for these two testing datasets are done independently, where we assume that the initial investment starts with 1 unit of liquid asset at the first day. In Figure 1, we demonstrate the convergence of the neural-progressive hedging method for four asset universes on the first testing dataset, in the presence of and after damping to zero the expert guidance after 20 iterations.

Figure 2(a)-(b) shows the mean and standard error in returns of NP with CVaR $\alpha = 0.95, 0.99$, along with the unconstrained RL policy and the pure SP policy, over four asset universes. The NP policies with CVaR $\alpha = 0.95, 0.99$ significantly outperforms the pure SP policy and improves the average return by 14% and 18% over the DDPG policy. It should be noted that the variance (demonstrated by the light shaded area) arises from differences in return rates for 4 different asset universes, but our NP method always outperforms other baseline methods for individual asset universe. Table 1 provides performance metrics including

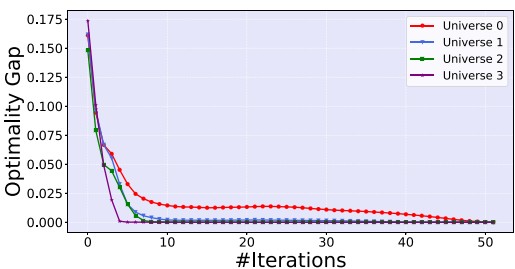

Figure 1: Convergence of the NP algorithm with CVaR$_{\alpha=95}$. According to Theorem 1, initial iterates may decrease non-monotonically but for iterations $i \geq \hat{i} = 20$ progression to an optimum is monotonic.

the Sharpe ratio, volatility and maximum daily drawdown (MDD), as well as the performance of four well-known and best performing online portfolio selection algorithms as benchmarks: (i) A uniform constant rebalancing portfolio (uCRP) approach [Cover, 2011]; (ii) Online moving average reversion (OLMAR) [Li and Hoi, 2012]; (iii) Passive-aggressive mean reversion (PAMR) [Li et al., 2012] and (iv) Robust median reversion (RMR) [Huang et al., 2016]. We use a grid search to optimize the two key hyper-parameters of these online universal portfolio algorithms: namely the lookback window $w$ and threshold parameter $\epsilon^3$. Average returns and Sharpe ratios of the NP are higher than all the benchmark approaches.

In Figure 2(c), we demonstrate the sample efficiency of our expert-guided NP approach. For this experiment, we train a DDPG policy with fewer samples (referred as "DDPG-LS", "LS=less samples") obtained after 0.5 million training steps, and use it as the expert policy to guide our NP approach. It should be noted that the data for generating scenario samples for the SP counterpart in our NP method is from the 0.5 million step samples used during training of sub-optimal DDPG only. Therefore, despite having less training data, NP still provides better returns than the sample-hungry DDPG policy, which is trained to convergence at 1.5 million steps.

A significant benefit of the NP framework is the ability to enforce constraints, otherwise difficult to handle in an RL policy. Figure 3(a)-(b) show the mean and standard error in returns under liquidity constraints. We compare against a heuristic we call DDPG-H that uses DDPG, but reserves $\mu_\ell + 3\sigma_\ell$ of the funds for the 0-interest cash account by re-normalizing the remaining allocations. DDPG-H thus provides a conservative, but constraint-feasible policy, by construction. The constrained NP policy outperforms CPO, PPO-L, DDPG-H and even the unconstrained DDPG policy.

In Figure 3(c), we show constraint violations; the red increasing line shows cumulative liquidity demand in each

---

[3] The source codes for the online portfolio selection algorithms can be found at https://github.com/Marigold/universal-portfolios.

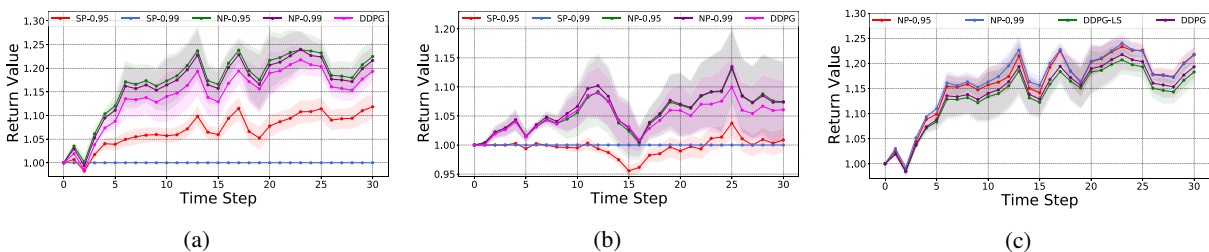

Figure 2: (a)–(b) Returns without liquidity constraints. NP policies outperform DDPG, SP; (c) Sample efficiency of NP.

| Algorithms | First 30 days, annualized values | | | | Second 30 days, annualized values | | | |
| | Returns | Sharpe | Volatility | MDD | Returns | Sharpe | Volatility | MDD |
|---|---|---|---|---|---|---|---|---|
| SP-0.0 | 11.84 | 3.58 | 27.33 | 7.63 | 2.3 | 1.16 | 17.8 | 4.8 |
| SP-0.95 | 11.83 | 3.58 | 27.37 | 7.65 | 0.87 | 0.51 | 17.32 | 4.85 |
| SP-0.99 | 0.0 | -0.54 | **0.0** | **0.0** | 0.0 | -3.78 | **0.0** | **0.0** |
| **NP-0.0** | **22.47** | 4.44 | 40.29 | 10.46 | **7.44** | **2.22** | 29.1 | 7.41 |
| **NP-0.95** | **22.47** | 4.44 | 40.29 | 10.46 | **7.44** | **2.22** | 29.1 | 7.41 |
| NP-0.99 | 21.64 | 4.29 | 40.38 | 10.44 | 7.4 | 2.09 | 30.96 | 7.99 |
| DDPG | 19.33 | 4.36 | 35.63 | 9.52 | 6.08 | 2.12 | 24.86 | 5.93 |
| uCRP | 12.08 | **5.68** | 17.16 | 5.26 | 1.38 | 0.97 | 12.5 | 3.77 |
| OLMAR | 10.4 | 4.65 | 18.26 | 5.97 | -4.17 | -2.69 | 12.98 | 3.54 |
| PAMR | 6.35 | 2.45 | 22.08 | 6.03 | -8.02 | -3.39 | 20.1 | 5.19 |
| RMR | 10.68 | 4.72 | 18.45 | 5.97 | -4.56 | -2.82 | 13.58 | 3.83 |

Table 1: Performance metrics without liquidity constraints. NP policies nearly always outperform all other strategies. SP with $\alpha = 0.99$ puts all funds in cash, hence MDD and volatility are 0, but returns are 0 as well.

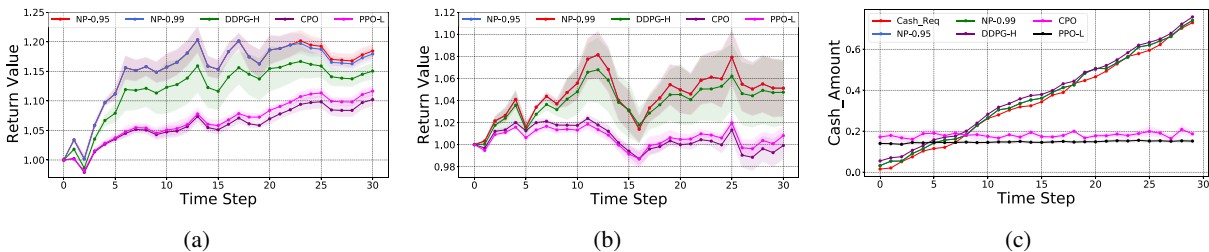

Figure 3: (a)-(b) Returns with liquidity constraints. NP with $\text{CVaR}_{\alpha=95}$ and $\text{CVaR}_{\alpha=99}$ are nearly identical in (b). NP policies outperform CPO, PPO-L and DDPG-H; (c) Average liquidity in each policy, Liquidity constraint is shown in red.

period. Although PPO-L, unlike CPO, was able to learn the constraints during training, both CPO and PPO-L are unable to come close to satisfying the liquidity constraints in testing (see the Supplementary Materials for constraint violations during training). Only DDPG-H and NP satisfy the constraints in testing, but the DDPG-H method over-allocates to the liquid account, thereby reducing net returns.

## ONLINE REPOSITIONING IN BIKE-SHARING

The bike repositioning problem is a form of online resource matching in an uncertain environment. Uncoordinated movements of users in bike or electric vehicle sharing, along with demand uncertainty, results in the need to often reposition the resources [Ghosh et al., 2017, Schui-

jbroek et al., 2017, Ghosh et al., 2016, Ghosh and Varakantham, 2017]. We use an RL-based simulator from Bhatia et al. [2019] built upon the dataset of Hubway bike sharing system in Boston, consisting of 95 base stations and 760 bikes. The state at time $t$ includes the current allocated bikes in each station $j \in \{1 \dots J\}$ and $\xi_{t,j}$ is the random customer demand at station $j$. The action is a vector, $x_t = \{x_{t,1}, ..., x_{t,J}\}$, that represents the percent allocations of bikes across all stations while respecting the constraint set $\mathcal{G}(\xi) := \{x | \check{C}_j \leq N \cdot x_{t,j} \leq \hat{C}_j, \sum_j x_{t,j} = 1, t = 1 \dots T\}$, where $\check{C}_j$ and $\hat{C}_j$ denote the minimum and maximum bounds on the number of allocated bikes at station $j$, and $N$ denotes the total number of bikes present in the system. The

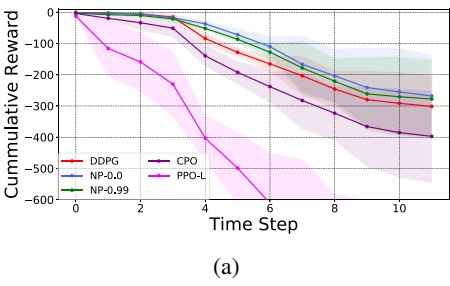 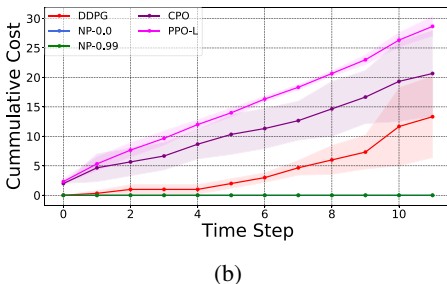

(a) (b)

Figure 4: Performance comparison on the online bike repositioning problem for 12 time steps (6AM-12PM), averaged over 3 testing days: (a) cumulative reward value; (b) cumulative constraint violation cost.

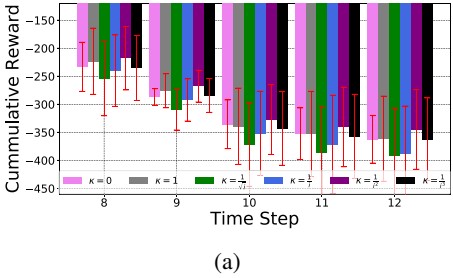 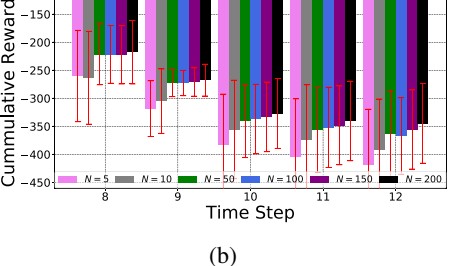

(a) (b)

Figure 5: Sensitivity analysis results by varying (a) convex combination parameter, $\kappa$; and (b) number of scenarios.

objective function is represented by:

$$\max_x \sum_t \sum_j -L(x_{t,j},\xi)(1+\log(1+L(x_{t,j},\xi))) - $$
$$R(x_{t,j},\xi)\sin(\pi \cdot R(x_{t,j},\xi))$$

where $L(x_{t,j},\xi)$ represents the amount of unfulfilled demand and $R(x_{t,j},\xi)$ is the number of bikes picked up or dropped off at station $j$ at time $t$ in action $x$, which incurs a repositioning cost. We use two months of data to train the models. We evaluate the learnt policies on 3 consecutive days during the morning peak period (6AM–12PM) with 12 decision epochs, each having a duration of 30 minutes.

Figure 4(a) shows mean and standard error in cumulative reward over 3 test days from NP with CVaR of $\alpha = 0.0, 0.99$, CPO and PPO-L, and unconstrained DDPG. After training CPO and PPO-L methods for 1 million episodes, they fail to perform at par with NP. NP using expected reward (i.e., $\alpha = 0.0$) and with CVAR of $\alpha = 0.99$ improves cumulative reward by 11.3% and 7.9% over unconstrained DDPG. Figure 4(b) shows the cumulative constraint violation where capacity violation per station costs 1 unit. Only NP variants satisfy the constraints, while both CPO and PPO-L fail to satisfy the capacity constraints during the test period.

Finally, we provide sensitivity analysis by varying (a) convex combination parameter, $\kappa$; and (b) number of scenarios sampled from the demand distribution. Figure 5(a) shows the mean and standard error in cumulative reward for different $\kappa$ that decreases over iterations of NP. Recall that at $\kappa = 1$ and 0, NP reduces to DDPG and pure SP, respectively.

The best performance is achieved with $\kappa = \frac{1}{i^2}$, which is our default setting in the experiments. Figure 5(b) shows the cumulative reward of NP in the last five time steps, varying the number of scenarios. As expected, the NP performance improves with the number of scenarios, as the approximate distribution and hence the scenario tree formulation tends towards the true distribution. However, the improvement happens in a concave manner and beyond some number of scenarios, the performance gain hits a plateau, implying that computation requirements remain reasonable to achieve a near-optimal solution with NP. We thus use 200 scenarios for both the pure SP and NP in the default settings of experiments.

## RELATED WORK

The neural-progressive hedging algorithm combines offline policy search with an online model-based phase to fine-tune the policy so as to satisfy constraints and risk measures such as CVaR. We categorize the existing relevant research into three threads: (a) Combining model-free and model-based methods for performance improvement; (b) Constrained and risk-sensitive RL methods; and (c) Improving sequential decisions through warm starting and imitation learning.

**Ensemble of model-free and model-based methods:** Model-based methods are prized for sample efficiency, but, as noted by Feinberg et al. [2018], high-capacity models are "prone to over-fitting in the low-data regimes where they are most needed", implying that the combination of

model-based and model-free methods will be important for good performance in complex settings. They propose, as do Buckman et al. [2018], to rollout the learned model for use in value estimation of a model-free RL, in the latter reference using an ensemble of such models to estimate variance. Ghosh et al. [2021] demonstrate the efficiency of combining the model-based and model-free RL methods in a complex air traffic control domain. Lu et al. [2019], Amos et al. [2018], Tamar et al. [2017], Kahn et al. [2017] combine (online) planning models with model-free RL to explore more sparingly the state and action spaces. Mansard et al. [2018] suggest a structure similar to ours for controlling dynamical systems using models to initialize a model predictive control formulation, as a warm-start. Lu et al. [2019] develop Adaptive Online Planning (AOP) with a continuous model-free RL algorithm, TF3 [Fujimoto et al., 2018]. The goal is similar to ours – leveraging the responsiveness of online planning with reactive off-policy learning to make better decisions. The approach is however different from ours – AOP uses a model-based policy when uncertainty is high and a reactive model-free policy when habitual behavior should suffice.

**Constrained and risk-sensitive RL methods:** García and Fernández [2015] surveyed safe RL methods which they classify as either optimization-based or handling safety in the exploration process. Pham et al. [2018] suggest after each policy update to project the current iterate onto the feasible set of safety constraints; since they assume that the safety constraints may not be known in advance, they propose a method to learn the parameters of a linear polytope. Yang et al. [2019] extend CPO method to solve constrained RL by optimizing the reward function using TRPO and then projecting the solution onto the feasible region defined by safety constraints, similar to Pham et al. [2018]. Chow et al. [2015, 2017] model risk-constrained MDPs with a CVaR objective or chance constraints, and solve it by relaxing the constraints and using a policy gradient algorithm. However, similar to CPO and Lagrangian-relaxed PPO, the constraints are not enforced during execution and need not be satisfied. Most "safe" RL methods use an initial infeasible, unconstrained policy and iteratively render it feasible and locally optimal, e.g., Berkenkamp et al. [2017] define an expanding "region of attraction" to guide safe exploration to improve the policy, whilst remaining feasible.

**Imitation learning and warm start:** NP can be viewed through the lens of imitation learning. Gu et al. [2016] use synthetic model-based "imagination" rollouts in the early iterations of deep RL training, which can be considered as a model-based warm-start. This is the opposite of our approach, we propose a longer-horizon deep RL to warm start the online stochastic program. Aggravate [Ross and Bagnell, 2014] and Aggravated [Sun et al., 2017], building on the seminal DAgger [Ross et al., 2011], involve iteratively mixing the learning step of a policy with an expert

policy, in that, at iteration $n$, $\pi^n = \beta^n \pi^* + (1 - \beta^n)\hat{\pi}^n$, where $\beta \to 0$ as $n \to \infty$. This is similar to the update step of NP which uses a convex combination of the expert and the learner policies, with damping. Cheng et al. [2018] take this one step further by defining a framework with a mirror descent gradient update that reduces to imitation learning-based RL, depending on the choice of the gradient estimator; they introduce SLOLS, where the gradient is a convex combination of a policy gradient and an expert gradient. Plato [Kahn et al., 2017] is similar with the roles of the expert and learner reversed: the Plato expert replans at each step to avoid catastrophic failure in training, while the learner is a neural network. Sun et al. [2018] propose combining imitation learning and RL with the aim of faster learning and improving beyond a sub-optimal expert. The advantages achieved by the NP method in inverting the roles of expert and learner are the ability of the SP to enforce hard constraints and incorporate risk measures, and doing so in an explainable manner. The NP warm start serves as an external expert to guide the SP in the early iterations to encourage convergence to a better solution by reshaping the objective itself.

## CONCLUSION

The neural-progressive hedging (NP) method starts from an offline, unconstrained RL policy and iteratively enforces constraints and risk requirements using model-based stochastic programming. It is thus a type of external point method. We demonstrate the efficacy of NP method on two real-world applications, in finance and logistics. The NP method significantly outperforms both constrained and unconstrained RL whilst handling both resource constraints and risk measures. An important benefit of the framework is its ease of implementation: NP method can be implemented using existing deep RL algorithms and commercial off-the-shelf optimization packages, and provides added transparency and explainability on the constraint satisfaction of the policy. One interesting direction for future work is the online re-formulation of the problem to take into account a time-varying distribution of $\xi$. Similarly, upper bounding the loss from such a re-solving policy would be of great interest.

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
