# OpenReview forum: "Neural-Progressive Hedging: Enforcing Constraints in Reinforcement Learning with Stochastic Programming"
_auai.org/UAI/2022/Conference — UAI 2022 Poster_

### Official Review · Reviewer_Jydc · 2022-04-06

**Q2(1) Originality/Novelty:** 3
**Q2(2) Significance/Impact:** 2
**Q2(3) Correctness/Technical Quality:** 3
**Q2(6) Clarity Of Writing:** 3
**Q6 Overall Score:** 6
**Q8 Confidence In Your Score:** 3

**Q1 Summary And Contributions:**

This paper studies the problem of learning an optimal policy in a Markov Decision Process (MDP) model which satisfies constraints such as conditional value-at risk (CVaR) during the execution of the policy. The proposed method leverages stochastic programming during the online phase of executing a reinforcement learning (RL) policy.

**Q2 Assessment Of The Paper:**

More detailed information regarding each of these aspects is given below:

**Q2(4) Quality Of Experiments (Optional):**

3: Good: The experimental evaluation is adequate, and the results convincingly support the main claims.

**Q2(5) Reproducibility:**

3: Good: Key resources (e.g., proofs, code, data) are available and key details (e.g., proofs, experimental setup) are sufficiently well-described for competent researchers to confidently reproduce the main results.

**Q3 Main Strengths:**

This paper studies an interesting problem. The problem of learning an optimal policy subject to some budget constraints seems relevant to many practical challenges, e.g., the class of resource allocation subject to structural constraints during policy execution.

The proposed method in Algorithm 1 appears to be technically sound. The key to Algorithm 1 replies on a novel offline RL optimization procedure. Theorem 1 ensures that it converges to the optimal solution under some reasonable convex assumptions. The authors performed extensive experiments. The simulation results supports the efficiency of the proposed approach.

**Q4 Main Weakness:**

The proposed algorithm is presented as an online RL algorithm. However, its exploration strategy seems a bit naive. It appears to be a greedy algorithm that always performs an action that appears the best from the empirical estimates. It has been known that the greedy strategy does not perform well in the online setting. I would like to see the online analysis of Algorithm 1, e.g., showing that its cumulative regret converges as the number of episodes T grows infinite.

**Q5 Detailed Comments To The Authors:**

Please see Q4.

**Q7 Justification For Your Score:**

This paper studies an interesting problem of sequential resource allocation with respect to typical resource constraints. It captures challenges in many practical applications including portfolio optimization with liquidity constraints for financial planning. The proposed method seems reasonable and could converge to the optimal solution. While the online regret analysis is missing, the offline RL component appears novel and could be useful for the AI community.

**Q9 Complying With Reviewing Instructions:**

1: Yes.

---

### Official Review · Reviewer_5brW · 2022-04-12

**Q2(1) Originality/Novelty:** 3
**Q2(2) Significance/Impact:** 3
**Q2(3) Correctness/Technical Quality:** 3
**Q2(6) Clarity Of Writing:** 3
**Q6 Overall Score:** 8
**Q8 Confidence In Your Score:** 1

**Q1 Summary And Contributions:**

This paper proposes a modification to existing RL methods where the goal is to
ensure that constraints are satisfied and also feasibility of risk measures.


**Q2 Assessment Of The Paper:**

More detailed information regarding each of these aspects is given below:

**Q2(4) Quality Of Experiments (Optional):**

3: Good: The experimental evaluation is adequate, and the results convincingly support the main claims.

**Q2(5) Reproducibility:**

3: Good: Key resources (e.g., proofs, code, data) are available and key details (e.g., proofs, experimental setup) are sufficiently well-described for competent researchers to confidently reproduce the main results.

**Q3 Main Strengths:**

I'm afraid that, as told to the conference organizers, this paper is far from my
field of expertise and thus, all I could do is to verify that there are no typos
and that it is seemingly well written.

As a foreigner, though, the paper looks interesting and I assume it is novel,
but I could not assess on the novelty or importance of the methods employed
here.


**Q4 Main Weakness:**

I'm afraid that, as told to the conference organizers, this paper is far from my
field of expertise and thus, all I could do is to verify that there are no typos
and that it is seemingly well written.

As a foreigner, though, the paper looks interesting and I assume it is novel,
but I could not assess on the novelty or importance of the methods employed
here.


**Q5 Detailed Comments To The Authors:**

I'm afraid that, as told to the conference organizers, this paper is far from my
field of expertise and thus, all I could do is to verify that there are no typos
and that it is seemingly well written.

As a foreigner, though, the paper looks interesting and I assume it is novel,
but I could not assess on the novelty or importance of the methods employed
here.


**Q7 Justification For Your Score:**

Just guessing, as I repeatedly said this paper falls very far from my field of expertise

**Q9 Complying With Reviewing Instructions:**

1: Yes.

---

### Official Review · Reviewer_Giw1 · 2022-04-13

**Q2(1) Originality/Novelty:** 3
**Q2(2) Significance/Impact:** 3
**Q2(3) Correctness/Technical Quality:** 3
**Q2(6) Clarity Of Writing:** 3
**Q6 Overall Score:** 8
**Q8 Confidence In Your Score:** 3

**Q1 Summary And Contributions:**

The paper proposed a framework, called neural-progressive hedging (NP), that leverages stochastic programming during the online phase of executing a reinforcement learning (RL) policy.

**Q2 Assessment Of The Paper:**

More detailed information regarding each of these aspects is given below:

**Q2(4) Quality Of Experiments (Optional):**

3: Good: The experimental evaluation is adequate, and the results convincingly support the main claims.

**Q2(5) Reproducibility:**

3: Good: Key resources (e.g., proofs, code, data) are available and key details (e.g., proofs, experimental setup) are sufficiently well-described for competent researchers to confidently reproduce the main results.

**Q3 Main Strengths:**

The paper is relevant to AI.

The conceptual approach seems to be technically sound, with no obvious flaws in it.

The claims that the proposed algorithm is an improvement over existing ones is supported by theory and experimental results.

The text is clear and well written. The paper is well organized, with well-structured sentences, where we can understand the development of the argument well. The abstract gives a good description of the paper, summarizing it well. Figures, diagrams and formulas are all readable and formatted correctly.


**Q4 Main Weakness:**

The research methodology could be improved – I do not think it is possible to reproduce the experiments using only the information in the paper (an important point in scientific research). But I am not sure there would be space for this information.

**Q5 Detailed Comments To The Authors:**

The approach to solve the proposed problem is novel, seems sound, and will have an impact in the field. It is well written, clear, and looks reproductible.

Therefore, the paper should be accepted.


**Q7 Justification For Your Score:**

A very theoretical paper, with very good results

**Q9 Complying With Reviewing Instructions:**

1: Yes.

---

### Official Review · Reviewer_uiQo · 2022-04-21

**Q2(1) Originality/Novelty:** 2
**Q2(2) Significance/Impact:** 2
**Q2(3) Correctness/Technical Quality:** 3
**Q2(6) Clarity Of Writing:** 3
**Q6 Overall Score:** 4
**Q8 Confidence In Your Score:** 3

**Q1 Summary And Contributions:**

This work focuses the resource allocation problem where its goal is to ensure feasibility with respect to constraints and risk-based objectives. Specifically, the neural-progressive hedging (NP) framework is proposed where it trains an unconstrained RL first, then a stochastic program is used to re-optimize the policy under constraints and risk measures during the online execution. Finally, experiments on two real-world tasks have been conducted, with results demonstrating its effectiveness.


**Q10 Ethical Concerns (Optional):**

Yes. This work targets to the real-world problems, however, there lacks the discussion on the concerns under risk-sensitive setting.


**Q2 Assessment Of The Paper:**

More detailed information regarding each of these aspects is given below:

**Q2(4) Quality Of Experiments (Optional):**

2: Fair: The experimental evaluation is weak: important baselines are missing, or the results do not adequately support the main claims.

**Q2(5) Reproducibility:**

2: Fair: Key resources (e.g., proofs, code, data) are unavailable but key details (e.g., proof sketches, experimental setup) are sufficiently well-described for an expert to confidently reproduce the main results.

**Q3 Main Strengths:**

- The resource allocation problem is closely related to the real-world applications, which would be interesting to both the theoretical and applied researchers in RL community. It is novel to propose the neural-progressive hedging framework to solve this resource allocation problem under constraints and risk measures.
- The methodology is generally sound to me. Besides, this paper also makes the theoretical analysis for the proposed method.


**Q4 Main Weakness:**

- In the section of introduction, this paper claimed that there are three shortcomings, which impede the uptake of RL for real-world problems. However, this claim only holds for the constrained or risk-sensitive RL problems rather than the "general" RL problems.
- The comparison for sample efficiency is unfair. It looks NP has an ensembled model for handling different scenarios while DDPG is often single model. Therefore, the comparison should be counted in terms of the total number of samples required for each model rather than the time steps.
- Current results didn't support the claim on better explainability of the policies. It looks the number of scenarios is more like a hyper-parameter.


**Q5 Detailed Comments To The Authors:**

- See my comments in the main weakness.
- -Both the method and the theoretical analysis are based on the assumption that $\tilde(f)$ is a convex function. However, there lacks the discussion on how to garantee the convexity under the real-world setting. Beside, NP relies on the learned model and assumes this model should be good approximations to the true state transitions. How to garantee a sufficiently good model under the real-world setting?
- What does the "scalable manner" in this paper refer to?
- What does the "unseen scenarios" in this paper refer to? Since NP relies on a good model, the transitions should be known and keep fixed. It looks to me that this unseen scenarios refer to the partially observable states, is it true?
- From Figure 4, it looks the more scenarios, the better performance is. Is this true?

**Q7 Justification For Your Score:**

Considering the current form of the paper, it feels to me that the main weaknesses outweigh the main strength a little bit. Therefore, I prefer to borderline reject.


**Q9 Complying With Reviewing Instructions:**

1: Yes.

---

### Decision · Program_Chairs · 2022-05-15

**Decision:**

Accept (Poster)

**Comment:**

Meta Review: The authors have provided a strong rebuttal, which they are encouraged to incorporate into their paper on revision to clarify concerns raised by reviewers.  While one reviewer retains their concerns after the rebuttal, three other reviewers side with acceptance.